# Does Elderly-Onset Inflammatory Bowel Disease Increase Risk of Colorectal Cancer? A Systematic Review and Meta-Analysis

**DOI:** 10.3390/jcm13010148

**Published:** 2023-12-27

**Authors:** Mohammad Zamani, Shaghayegh Alizadeh-Tabari

**Affiliations:** Digestive Diseases Research Center, Digestive Diseases Research Institute, Tehran University of Medical Sciences, Tehran 1411713135, Iran

**Keywords:** inflammatory bowel disease, elderly, colorectal cancer, systematic review

## Abstract

Background: Although younger adults with inflammatory bowel disease (IBD) are known to have an increased risk of developing colorectal cancer (CRC), the impact of IBD on CRC risk in elderly patients is not yet fully understood. Therefore, we conducted this systematic review and meta-analysis to address this knowledge gap. Methods: We thoroughly searched Embase, PubMed, and Scopus, covering the literature from inception to 31 August 2023, in any language. We enrolled population-based cohort studies that appraised the risk of CRC development in elderly patients (≥60 years) with IBD as compared to the non-IBD population. Our meta-analysis provided pooled relative risk (RR) with 95% confidence intervals (CIs) using a random-effect model. Results: Out of 3904 citations, 3 eligible cohort studies were ultimately included, reporting 694 CRC events in 35,187 patients with IBD. According to analysis, the risk of developing CRC did not increase in patients with elderly-onset IBD (RR = 1.17 [95% CI: 0.86–1.47]; I^2^ = 62.6%). This lack of a significant association was observed in both patients with Crohn’s disease (RR = 1.28 [95% CI: 0.88–1.69]) and ulcerative colitis (RR = 0.99 [95% CI: 0.90–1.09]) (*p* for interaction = 0.166). Conclusion: Our findings revealed no significant increase in the risk of incident CRC in patients with elderly-onset IBD, suggesting that intense screening of these patients for CRC may not be necessary.

## 1. Introduction

Inflammatory bowel disease (IBD) is a long-lasting and incapacitating disorder that involves inflammation of the digestive tract, mainly including Crohn’s disease (CD) and ulcerative colitis (UC) [1]. IBD affects millions of people worldwide, with a higher incidence in developed countries and a rising prevalence globally [2]. The cause of IBD is not clearly understood, but it is thought to be the result of a complex interplay between genetic predisposition, environmental factors, and an abnormal immune response to gut microbiota [3,4]. The pathophysiology of IBD involves an abnormal and excessive immune response in the intestinal mucosa [5]. The manifestations of IBD extend beyond the intestinal tract, affecting various organ systems. Intestinal symptoms include abdominal pain, diarrhea, rectal bleeding, and weight loss [6]. Extra-intestinal manifestations involve the joints, skin, eyes, and liver, leading to conditions such as arthritis, erythema nodosum, uveitis, and primary sclerosing cholangitis [7]. Additionally, patients with IBD may experience systemic symptoms, such as fatigue and fever [8].

IBD can also be associated with incr eased risks of gastrointestinal malignancies, such as colorectal cancer (CRC), which is one of the most common causes of cancer-related mortality around the world [9,10]. However, the relationship between IBD and CRC is complex and multifaceted, with numerous factors influencing the risk. A crucial aspect of this association that has garnered limited attention is the age of disease onset. IBD is often diagnosed in young adulthood or early middle age. However, a notable subset of patients experiences disease onset in their later years, known as elderly-onset IBD (≥60 years old) [11,12]. The clinical presentation, course, and prognosis of elderly-onset IBD may differ from those of younger-onset cases due to the unique challenges associated with aging, comorbidities, and immunosenescence [13]. Although the risk of CRC incidence in younger adult patients with IBD is well understood [9,14], the impact of elderly-onset IBD on CRC risk remains inadequately characterized. Therefore, an inclusive study is needed to address this significant knowledge gap.

Understanding the relationship between elderly-onset IBD and CRC risk is particularly pertinent given the aging population demographic across the globe. Aging is a multifaceted biological process that leads to various physiological changes, including alterations in the immune system and increased susceptibility to chronic inflammatory conditions [15]. These age-related changes may interact with the pathophysiological mechanisms of IBD, potentially influencing the development of CRC. Additionally, older adults often experience delays in an IBD diagnosis due to atypical presentations and overlapping symptoms with other age-related gastrointestinal disorders [16]. Such delays may impact the disease severity and subsequent cancer risk, emphasizing the importance of a timely diagnosis and management in this population.

This systematic review and meta-analysis examined the existing literature to assess the risk of CRC in elderly-onset IBD comprehensively, shedding light on a critical yet underexplored dimension of this relationship. Given the increasing prevalence of elderly individuals living with IBD, this research has the potential to influence clinical practice and inform more tailored approaches to patient care, ultimately improving the quality of life and outcomes for these patients.

## 2. Materials and Methods

### 2.1. Information Sources and Search Strategy

We followed the instructions outlined by the Preferred Reporting Items for Systematic Review and Meta-Analysis (PRISMA) guidelines [17]. We conducted a thorough search across various scientific databases, including Embase, PubMed, and Scopus. Our search included papers published from inception to 31 August 2023, in any language, involving the following set of keywords applied to title/abstract: *Inflammatory bowel disease* OR *Inflammatory bowel diseases* OR *IBD* OR *Crohn* OR *Crohn’s disease* OR *Colitis* OR *Ulcerative colitis* AND *Colorectal* OR Colon OR *Rectum* OR *Rectal* AND *Cancer* OR *Cancers* OR *Malignancy* OR *Malignancies* OR *Carcinoma* OR *Adenocarcinoma* OR *Neoplasia* AND *Cohort* OR *Population-based* OR *Community-based*. To manage the references identified, we used EndNote 8.1 software (Clarivate Analytics, Philadelphia, PA, USA). This software helped us to organize the vast amount of data we gathered, making it easier to keep track of the sources we found. Moreover, we manually searched the reference lists of the pertinent reviews and the articles ultimately captured (grey literature) to avoid missing any relevant sources. As a result, we gained a deep understanding of the research landscape. We did not primarily register the protocol of our systematic review and meta-analysis in an online registry.

### 2.2. Inclusion and Exclusion Criteria

To ensure that we collected the most reliable and relevant information, we defined specific criteria for the selection of research studies. To be eligible, the research had to be retrospective or prospective cohort studies that were performed on at least 100 patients with elderly-onset IBD (≥60 years) without a prior CRC diagnosis when they first participated. IBD had to be diagnosed with histological or radiological tests. Additionally, new cases of incident CRC had to be reported after the diagnosis of IBD. Finally, the studies could provide any statistical measurements of association (compared with healthy populations) with their corresponding 95% confidence intervals (CIs), which had to be adjusted for potential confounding factors (at least sex and age). On the other hand, we excluded papers if they were reviews, case reports, editorials, letters to the editor, published duplicates, or studies investigating the same sample groups as well as those lacking clear information about their methodology or results related to the study outcome. Overall, we tried to take a comprehensive approach to identify the relevant research studies that met our eligibility criteria.

### 2.3. Study Selection and Data Extraction

We initially screened the references by reviewing their titles and abstracts, then evaluated the full texts with eligibility forms. Ultimately, we extracted the necessary information onto a Microsoft Excel spreadsheet (Microsoft Corporation, Redmond, WA, USA) after enrolling all suitable articles. Disagreements were resolved through consensus between the reviewers, with the agreement level measured using the kappa statistic. The data collected from the studies were as follows: first author’s name, publication year, study location, study period, study design, patients’ sample size, number of patients by sex, mean age, mean follow-up duration, number of CRC events, the risk estimates with 95% CIs, data on IBD type, and covariates used for adjustment. We considered it a separate report when multiple measures of association were provided for the same population. We used Google Translate for translating surveys in a language other than English. Concerning the duplicate publications, the most comprehensive and detailed one was selected.

### 2.4. Risk of Bias Assessment

The quality of the cohort studies included was appraised utilizing the Newcastle–Ottawa scale [18]. This tool examines various aspects of the studies, including selection of the study groups (with a maximum score of four in four domains), comparability of the groups (up to two scores), and ascertainment of the outcomes (up to three scores in three domains). The highest score achievable on this scale is 9, exhibiting studies with the lowest risk of bias and highest quality. Studies that received a score of 7 or higher were classified as ‘high quality’ [19]. The authors (M.Z. and S.A.T.) contributed to the quality assessment, and any discrepancies were resolved by consensus.

### 2.5. Study Outcome and Statistical Analysis

The primary outcome of interest was the CRC development throughout the follow-up period. We used a random-effect model to combine the risk estimates, resulting in more conservative overall pooled relative risks (RRs). We measured the heterogeneity between the studies using I-squared and chi-squared tests. In the chi-squared test, a *p*-value of less than 0.10 showed a significant heterogeneity. According to the I-squared values, we categorized the heterogeneity into low (25–49%), moderate (50–74%), or high (≥75%) [20]. We performed subgroup analysis based on IBD type (CD and UC) and study region (Europe and North America). A *p*-value less than 0.10 indicated a significant difference between subgroups (*p* for interaction) [21]. We chose a *p*-value threshold of 0.10 for subgroup analyses to balance mitigating type I errors and capturing potentially meaningful trends within specific subgroups. This slightly relaxed threshold increases sensitivity in identifying subgroup effects while maintaining a level of statistical stringency. A *p*-value < 0.10 suggests partial explanation of heterogeneity through stratification. All these analyses were done utilizing the STATA statistical software version 14.2 (StataCorp, College Station, TX, USA). This software enabled us to analyze data efficiently and draw meaningful conclusions based on the most recent and reliable scientific evidence.

## 3. Results

### 3.1. Search Results, Study Selection, and Characteristics

Out of 3904 citations identified through the initial database search, 1533 were excluded due to duplication. After screening the title/abstract or full texts of the remaining 2371 sources, we identified 3 eligible articles by excluding 2368 publications due to unsuitability [22,23,24]. Figure 1 illustrates the search strategy and study selection process in accordance with the PRISMA flow diagram [17]. The level of agreement between the reviewers was excellent, with a Kappa statistic of 0.92. The chosen articles were published in the English language and between 2016 and 2022. Out of the three studies retrieved, two were conducted in multiple centers and one in a single center. The studies were carried out in Canada (n = 1), France (n = 1), and multiple locations (n = 1). The median follow-up varied between 5 and 6 years. Two studies adjusted the risk estimate for sex and age only, and another study also adjusted for age of IBD diagnosis and country. All three studies were of a high quality. Table 1 and Table 2 represent the baseline characteristics and the risk of bias investigation of the included studies, respectively.

### 3.2. Overall Risk of Colorectal Cancer

There were three cohort studies that investigated the link between elderly-onset IBD and the risk of CRC occurrence, reporting 694 CRC events in 35,187 patients with IBD (Table 1). Our analysis showed that the risk of incident CRC did not increase in patients with elderly-onset IBD (pooled RR = 1.17 [95% CI: 0.86–1.47]), with moderate heterogeneity between the studies (I^2^ = 62.6%, *p* for χ^2^ = 0.069) (Figure 2). This insignificant association was observed in both patients with CD (pooled RR = 1.28 [95% CI: 0.88–1.69]; I^2^ = 38.9%, *p* for χ^2^ = 0.195) and UC (pooled RR=0.99 [95% CI: 0.90–1.09]; I^2^ = 0.0%, *p* for χ^2^ = 0.466) (*p* for interaction = 0.166). A subgroup analysis by study region showed a significant difference in the risk of CRC between studies conducted in Europe (2 studies, pooled RR = 1.03 [95% CI: 0.90–1.16]; I^2^ = 0.0%, *p* for χ^2^ = 0.999) and in North America (1 study, RR = 1.48 [95% CI: 1.16–1.88]) (*p* for interaction = 0.009). However, caution should be taken when comparing the subgroups due to the limited number of studies carried out.

Among the included studies, only Kuenzig et al. [24] reported a significantly raised risk of incident CRC in patients with elderly-onset IBD (RR = 1.48 [95% CI: 1.16–1.88]). On the other hand, based on their study, the magnitude of increased CRC risk was not significantly different between patients with CD (RR = 1.80 [95% CI: 1.22–2.65]) and those with UC (RR = 1.25 [95% CI: 0.90–1.75]) (*p* for interaction = 0.162).

Out of the studies we enrolled, Everhov et al. [23] maximally adjusted the CRC risk estimates for the covariates. However, the increased risk of CRC incidence in patients with elderly-onset IBD remained insignificant (RR = 1.03 [95% CI: 0.91–1.17]). Moreover, according to Everhov et al.’s study [23], when separately assessing the risks of colon and rectal cancers, it was found that elderly patients with IBD did not have a significant increase in the risk of either cancer (colon, RR = 1.37 [95% CI: 0.76–1.98]; rectum, RR = 0.91 [95% CI: 0.64–1.17]) (*p* for interaction = 0.156). On the other hand, the sex-specific results showed that the risk of developing CRC was significantly increased in women (RR = 1.41 [95% CI: 1.07–1.76]) compared with men (RR = 1.00 [95% CI: 0.68–1.32]) (*p* for interaction = 0.098).

## 4. Discussion

In this study, we endeavored to elucidate the unclear link between elderly-onset IBD and the risk of CRC development by reviewing the existing literature. After screening thousands of citations identified using rigorous eligibility criteria, we ultimately included three cohort studies comprising more than 35,000 patients with IBD. Based on the analysis, there was no significant increase in the CRC risk for those with elderly-onset IBD. Our study also found no significant association between elderly-onset IBD and CRC occurrence in patients with CD and UC.

Although there were few review articles previously published on the implications of IBD in elderly patients [25,26], to the best of our knowledge, our study is the first systematic review and meta-analysis specifically answering the query of whether elderly-onset IBD potentially increases the CRC risk. Prior meta-analyses included the younger population as well, demonstrating a direct connection between IBD and the risk of developing CRC. The meta-analysis by Lutgens et al. [14], enrolling population-based surveys, indicated that patients with IBD were significantly at a higher risk of CRC versus the non-IBD population (pooled standardized incidence ratio = 1.7 [95% CI: 1.3–2.1]). Similarly, in their study, Wan et al. [9] reported an increased CRC risk in patients with IBD (pooled odds ratio [OR] = 1.69 [95% CI: 1.46–1.91]) as well as in CD (OR = 1.47 [95% CI: 1.32–1.61]) and in UC (OR = 1.51 [95% CI: 1.28–1.74]). In addition to those reviews, the cohort research by Olén et al. indicated that patients with CD were at an increased risk of CRC mortality in any age group [27]. In contrast, patients with elderly-onset UC do not experience the same risk as other age groups [28]. However, these data are limited and cannot be relied upon for clinical practice. Further studies are necessary to clarify these associations.

The lack of a significant increase in the risk of incident CRC among elderly-onset patients with IBD could be attributed to various factors. First, the management and treatment of IBD have significantly improved over the years. Advancements in medical therapies, including immunomodulators and biological agents, have allowed for better control of inflammation in the gastrointestinal tract [29,30]. These treatments could not only help manage IBD symptoms but also reduce the risk of CRC development. Regular monitoring and early intervention can effectively prevent the progression of inflammation to cancerous stages, especially in elderly patients. Second, elderly-onset IBD patients are diagnosed at an older age when compared to younger individuals. The duration of IBD is an essential factor in the development of CRC. Long-standing inflammation in the colon and rectum can increase the risk of CRC [31]. As elderly patients are diagnosed later in life, they might have a shorter duration of inflammation compared to those diagnosed at a younger age, reducing their overall risk of developing CRC. Additionally, it has been noted that the immune response in elderly individuals differs from that in younger patients. The immune system in the elderly may have a decreased pro-inflammatory response, which could potentially attenuate the inflammatory processes associated with IBD. This altered immune response might contribute to a milder disease course in elderly patients, reducing the risk of CRC development [32,33]. Furthermore, lifestyle factors, such as smoking and diet, play a significant role in the development of CRC [34]. Studies have shown that elderly individuals tend to adopt healthier lifestyles, including quitting smoking and consuming a balanced diet [35,36], which can mitigate the risk of CRC. Additionally, regular physical activity, which is often encouraged in the elderly population, can contribute to a healthier colon and reduce the risk of cancer development. Moreover, advancements in surveillance techniques, such as colonoscopies and other screening methods, have enabled the early detection and removal of precancerous lesions, reducing the incidence of CRC in elderly-onset IBD patients [37]. Finally, different disease characteristics and potential genetic/environmental factors in elderly-onset IBD might also influence the risk of CRC development [25,38]. However, further research is needed to fully understand these complexities.

The results of this systematic review and meta-analysis carry significant implications for both patients and healthcare providers. First and foremost, these findings provide reassurance to elderly patients diagnosed with IBD. CRC is a serious concern for patients with IBD due to chronic inflammation and increased cell turnover in the colon and rectum. Patients, especially those in older age groups, often live with the fear of developing cancer. The study findings can alleviate anxiety among elderly patients with IBD, offering them comfort in knowing that their condition does not necessarily elevate their risk of developing CRC. For healthcare providers, these results have important implications for patient management and counseling. Clinicians can use this information to tailor their discussions with elderly patients newly diagnosed with IBD, emphasizing the importance of managing the disease for symptom control and quality of life rather than focusing solely on cancer risk. Healthcare providers can also use these data to guide their surveillance strategies. Regular screenings for CRC, such as colonoscopies and fecal occult blood tests, are standard practice for cases with IBD, as they are at an increased risk of developing CRC. However, these findings suggest that elderly-onset IBD patients may not need to undergo more aggressive or frequent screenings than the general population. This can help in optimizing healthcare resources and reducing the burden on both patients and the healthcare system. Moreover, these results underline the complexity of IBD and its various manifestations across different age groups. Understanding that the association between IBD and CRC risk varies based on age can inform future research endeavors. Researchers can explore the underlying mechanisms that might protect elderly patients with IBD from an increased cancer risk, potentially leading to insights that could be applicable to cancer prevention strategies for the general population. Finally, these findings highlight the importance of considering the heterogeneity within the IBD patient population. IBD is a multifaceted condition, and different subgroups of patients may have distinct disease trajectories and risks. Tailoring treatment and surveillance strategies based on age, disease onset, and other demographic factors could lead to more personalized and effective approaches to managing IBD and its associated complications.

A significant limitation of our systematic review and meta-analysis is the small number of included studies. This constraint hindered our ability to conduct a comprehensive assessment of publication bias and impeded the performance of subgroup analyses based on sex and colorectal cancer (CRC) location. The scarcity of studies, particularly in the context of indeterminate colitis [39], underscores the challenge of drawing definitive conclusions within this specific subgroup. A future direction for research is expanding the pool of available studies to enhance the robustness of analyses and enable more nuanced subgroup investigations. Another noteworthy limitation was the presence of moderate heterogeneity among the studies, particularly regarding the primary study outcome. While a sub-group analysis by inflammatory bowel disease (IBD) type partially justified this heterogeneity, it is essential to recognize that factors contributing to variation in study results may still exist. It is important to explore the sources of heterogeneity more in depth; this can be done by identifying additional subgroups or investigating other study characteristics that may influence outcomes. In addition, the analysis may not have fully accounted for confounding variables, including comorbidities, medications, and lifestyle factors, such as diet and habits (e.g., red and processed meat consumption, obesity, tobacco use, and alcohol use). The failure to address these variables comprehensively could impact the validity of our results. Addressing this limitation necessitates future research incorporating a more extensive set of covariates, employing advanced statistical methods or study designs to control potential confounding factors. Understanding the interplay between these variables and their influence on the relationship between elderly-onset IBD and CRC risk is crucial for providing more accurate and reliable conclusions. Collectively, these limitations emphasize the importance of approaching the interpretation of our findings with caution. The gaps in our knowledge underscore the need for additional research to understand the relationship between elderly-onset IBD and CRC risk comprehensively. Future studies should address the identified limitations, expand the scope of the investigation, increase the number of included studies, refine subgroup analyses, and account for a broader range of potential confounding variables. Only through such concerted efforts can we attain a more nuanced and accurate understanding of the complex interplay between elderly-onset IBD and the risk of colorectal cancer.

## 5. Conclusions

The findings of this systematic review and meta-analysis revealed no significant increase in the risk of incident CRC in patients with elderly-onset IBD. This lack of a significant association was also consistent for both patients with CD and UC patients. Thus, an intense screening of elderly patients with new IBD onset may not be necessary for CRC. However, further research is needed to confirm these results.

## Figures and Tables

**Figure 1 jcm-13-00148-f001:**
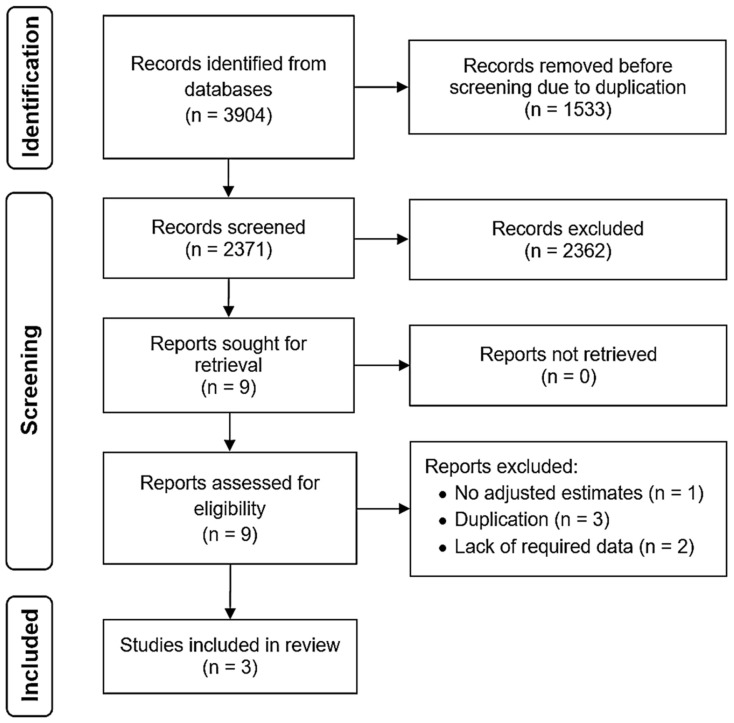
PRISMA flow diagram.

**Figure 2 jcm-13-00148-f002:**
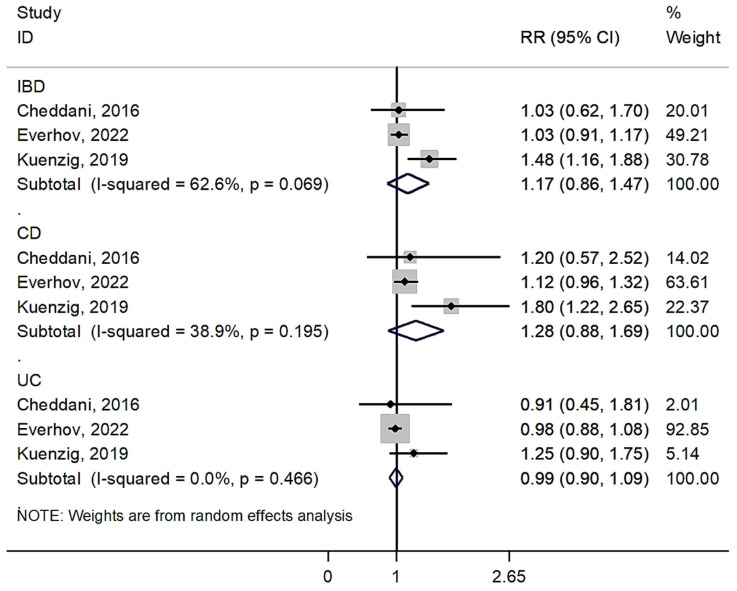
Forest plot of pooled relative risk (RR) of colorectal cancer with 95% confidence interval (CI) in patients with elderly-onset inflammatory bowel disease (IBD), Crohn’s disease (CD), and ulcerative colitis (UC).

**Table 1 jcm-13-00148-t001:** Basic characteristics of the studies included in the systematic review.

Study	Country	Study Period	Study Design	Patients with Inflammatory Bowel Disease					Adjustment for
				Total Patients (n)	Men (%)	Median Age (Years)	Median Follow-Up (Years)	CRC Events (n)	
**Cheddani, 2016**	France	1990–2006	Multi-center	IBD = 844 CD = 370 UC = 474	51.5	70	6	15	Sex and age
**Everhov, 2022**	Sweden and Denmark	1969–2017	Multi-center	IBD = 29,093 CD = 7869 UC = 21,224	47.4	69	6	569	Sex, age at IBD diagnosis, birth year, country
**Kuenzig, 2019**	Canada	2002–2013	Single-center	IBD = 5250 CD = 1683 UC = 3195	NA	NA	5	110	Sex and age

CRC: colorectal cancer, IBD: inflammatory bowel disease, CD: Crohn’s disease, UC: ulcerative colitis, NA: not available.

**Table 2 jcm-13-00148-t002:** Newcastle–Ottawa scale quality assessment of cohort studies investigating the association between elderly-onset inflammatory bowel disease and the risk of colorectal cancer.

Study	Selection (Out of 4)	Comparability (Out of 2)	Outcome (Out of 3)	Total (Out of 9)
**Cheddani, 2016**	4	1	3	8
**Everhov, 2022**	4	1	3	8
**Kuenzig, 2019**	3	1	3	7

## Data Availability

No additional data available.

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
