# Peer review of "Does Elderly-Onset Inflammatory Bowel Disease Increase Risk of Colorectal Cancer? A Systematic Review and Meta-Analysis"

_jcm, 2023, doi:10.3390/jcm13010148_

Round 1
Reviewer 1 Report
Comments and Suggestions for Authors
This systematic review and meta-analysis found no significant increase in the risk of colorectal cancer (CRC) in elderly-onset inflammatory bowel disease (IBD) patients, indicating that intensive CRC screening for this group may not be required. This study addresses a gap in knowledge regarding the risk of colorectal cancer (CRC) in elderly patients with inflammatory bowel disease (IBD), which can inform screening practices and healthcare policies.
However, here are some suggestions for the authors.
2.4. Risk of Bias Assessment
· It is commendable that a risk of bias assessment is being conducted. However, it is crucial to delineate if multiple reviewers will be involved in this process, and if so, how any discrepancies in scoring will be reconciled. This is a pivotal step in ensuring the objectivity and reliability of the bias assessment. A standardized approach or a consensus mechanism should be clearly described.
2.5. Study Outcome and Statistical Analysis
· The use of a p-value less than 0.10 to denote significance in subgroup analyses deviates from the conventional threshold of 0.05. It is essential to provide a rationale for this choice to ensure methodological rigor and to help readers understand the implications of this level of significance.
· While stating that the use of STATA for statistical analyses is beneficial, it would be advisable to specify the version used. Different versions of STATA may have variations in functionalities or algorithms, which can impact the results. Clarity on this aspect will allow for better reproducibility and comprehension of the analyses performed.
Basic Characteristics of Included Studies
· The omission of diet and lifestyle risk factors, notably the intake of red and processed meats, obesity, tobacco use, and alcohol use, is a significant oversight given their established roles in IBD and CRC. If these factors were not included in the selected studies, it is imperative to acknowledge this as a limitation in the study design, as their exclusion may affect the validity and applicability of the findings.
Discussion
· The claim that this is the first systematic review and meta-analysis to investigate the relationship between elderly-onset IBD and increased CRC risk appears to be inaccurate. A literature search has yielded several prior reviews on similar topics, indicating that this is not an unprecedented area of study. It is essential to accurately contextualize the study within the existing literature to maintain credibility and contribute meaningfully to the scientific discourse.
1. Everhov et al. "Colorectal cancer in elderly‐onset inflammatory bowel disease: a 1969–2017 Scandinavian register‐based cohort study." Alimentary Pharmacology & Therapeutics (2022).
2. Cheddani et al. "Cancer in elderly onset inflammatory bowel disease: a population-based study." Official journal of the American College of Gastroenterology (2016).
3. Gisbert and Chaparro. "Systematic review with meta‐analysis: inflammatory bowel disease in the elderly." Alimentary pharmacology & therapeutics (2014).
4. Ananthakrishnan et al. "AGA clinical practice update on management of inflammatory bowel disease in elderly patients: expert review." Gastroenterology (2021).
· These references indicate that prior studies have explored aspects of the topic in question, and thus, the discussion should reflect a more nuanced understanding of the scope of existing research.
Author Response
Reviewer 1:
This systematic review and meta-analysis found no significant increase in the risk of colorectal cancer (CRC) in elderly-onset inflammatory bowel disease (IBD) patients, indicating that intensive CRC screening for this group may not be required. This study addresses a gap in knowledge regarding the risk of colorectal cancer (CRC) in elderly patients with inflammatory bowel disease (IBD), which can inform screening practices and healthcare policies.
However, here are some suggestions for the authors.
2.4. Risk of Bias Assessment
- It is commendable that a risk of bias assessment is being conducted. However, it is crucial to delineate if multiple reviewers will be involved in this process, and if so, how any discrepancies in scoring will be reconciled. This is a pivotal step in ensuring the objectivity and reliability of the bias assessment. A standardized approach or a consensus mechanism should be clearly described.
Response: Thank you for your comment. We have added that the authors (M.Z. and S.A.T.) contributed to the quality assessment, and any discrepancies were resolved by consensus.
2.5. Study Outcome and Statistical Analysis
- The use of a p-value less than 0.10 to denote significance in subgroup analyses deviates from the conventional threshold of 0.05. It is essential to provide a rationale for this choice to ensure methodological rigor and to help readers understand the implications of this level of significance.
Response: We appreciate the reviewer for mentioning this point. We chose a p-value threshold of 0.10 for subgroup analyses to balance mitigating type I errors and capturing potentially meaningful trends within specific subgroups. This slightly relaxed threshold increases sensitivity in identifying subgroup effects while maintaining a level of statistical stringency. We have also stated that a p-value less than 0.10 suggests that stratifying based on a specific study characteristic partially explained the heterogeneity observed in the analysis. We have cited a reference regarding this part (ref 19).
- While stating that the use of STATA for statistical analyses is beneficial, it would be advisable to specify the version used. Different versions of STATA may have variations in functionalities or algorithms, which can impact the results. Clarity on this aspect will allow for better reproducibility and comprehension of the analyses performed.
Response: With thanks, we have added the STATA version 14.2.
Basic Characteristics of Included Studies
- The omission of diet and lifestyle risk factors, notably the intake of red and processed meats, obesity, tobacco use, and alcohol use, is a significant oversight given their established roles in IBD and CRC. If these factors were not included in the selected studies, it is imperative to acknowledge this as a limitation in the study design, as their exclusion may affect the validity and applicability of the findings.
Response: Thank you for your comment. We should clarify that we did not exclude those factors; however, no studies adjusted their analysis for those factors, and we have acknowledged it in our limitations (p 8).
Discussion
- The claim that this is the first systematic review and meta-analysis to investigate the relationship between elderly-onset IBD and increased CRC risk appears to be inaccurate. A literature search has yielded several prior reviews on similar topics, indicating that this is not an unprecedented area of study. It is essential to accurately contextualize the study within the existing literature to maintain credibility and contribute meaningfully to the scientific discourse.
- Everhov et al. "Colorectal cancer in elderly‐onset inflammatory bowel disease: a 1969–2017 Scandinavian register‐based cohort study." Alimentary Pharmacology & Therapeutics (2022).
- Cheddani et al. "Cancer in elderly onset inflammatory bowel disease: a population-based study." Official journal of the American College of Gastroenterology (2016).
- Gisbert and Chaparro. "Systematic review with meta‐analysis: inflammatory bowel disease in the elderly." Alimentary pharmacology & therapeutics (2014).
- Ananthakrishnan et al. "AGA clinical practice update on management of inflammatory bowel disease in elderly patients: expert review." Gastroenterology (2021).
- These references indicate that prior studies have explored aspects of the topic in question, and thus, the discussion should reflect a more nuanced understanding of the scope of existing research.
Response: With thanks, the studies by Everhov et al. and Cheddani et al. were original research articles (but not review articles) that were included in our meta-analysis. The reviews by Gisbert and Chaparro, as well as Ananthakrishnan et al., did not specifically focus on the risk of CRC following elderly-onset IBD; however, we cited these references to our manuscript (refs 24 and 25) and clarified the issue raised by you.
Reviewer 2 Report
Comments and Suggestions for Authors
Majority of research of colorectal CA in IBD has focused on young onset patients. Now new research has focused on elderly-onset IBD and whether or not it is related to an increased incidence and mortality rate of colorectal cancer. This is a very timely meta-analysis that is essential to guide clinical guideline policy stratification for elderly onset IBD. My comments are as follows:
1. With respect to the types of IBD being adjudicated for risk of colorectal cancer, only the Crohn's Disease and Ulcerative Colitis were included in the study. Indeterminate colitis is a subtype of IBD accounting for 5-15% of patients with IBD. There has also been a case series published which warrants justification for exclusion of this IBD subtype from this meta-analysis. (Bernardino C. Branco, Noam Harpaz, David B. Sachar, Alexander J. Greenstein, Parissa Tabrizian, Joel J. Bauer, Adrian J. Greenstein, Colorectal Carcinoma in Indeterminate Colitis, Inflammatory Bowel Diseases, Volume 15, Issue 7, 1 July 2009, Pages 1076–1081, https://doi.org/10.1002/ibd.20865).
2. Please consider inclusion of these studies to further strengthen your pooling:
A. https://www.thelancet.com/journals/lancet/article/PIIS0140-6736(19)32545-0/fulltext
B.https://journals.lww.com/ajg/abstract/2016/10000/cancer_in_elderly_onset_inflammatory_bowel.19.aspx
Author Response
Reviewer 2:
Majority of research of colorectal CA in IBD has focused on young onset patients. Now new research has focused on elderly-onset IBD and whether or not it is related to an increased incidence and mortality rate of colorectal cancer. This is a very timely meta-analysis that is essential to guide clinical guideline policy stratification for elderly onset IBD. My comments are as follows:
- With respect to the types of IBD being adjudicated for risk of colorectal cancer, only the Crohn's Disease and Ulcerative Colitis were included in the study. Indeterminate colitis is a subtype of IBD accounting for 5-15% of patients with IBD. There has also been a case series published which warrants justification for exclusion of this IBD subtype from this meta-analysis. (Bernardino C. Branco, Noam Harpaz, David B. Sachar, Alexander J. Greenstein, Parissa Tabrizian, Joel J. Bauer, Adrian J. Greenstein, Colorectal Carcinoma in Indeterminate Colitis, Inflammatory Bowel Diseases, Volume 15, Issue 7, 1 July 2009, Pages 1076–1081, https://doi.org/10.1002/ibd.20865).
Response: Thank you for your comment. We did not restrict IBD type to only CD or UC. However, no studies were found on the indeterminate colitis, which has been added to our limitations (p 7).
- Please consider inclusion of these studies to further strengthen your pooling:
- https://www.thelancet.com/journals/lancet/article/PIIS0140-6736(19)32545-0/fulltext
B.https://journals.lww.com/ajg/abstract/2016/10000/cancer_in_elderly_onset_inflammatory_bowel.19.aspx
Response: With thanks, the article written by Olén et al. duplicates the study conducted by Everhov et al., which is already included in our meta-analysis (ref 22). Since Everhov et al.'s study has more comprehensive data, we have included it instead. Furthermore, we have also included Cheddani et al.'s study in our meta-analysis (ref 21).